# The Interestingness of Fonts

Category: Research

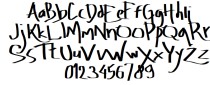 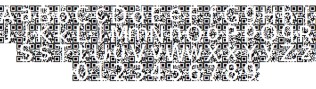

Figure 1: The top row shows the 5 most interesting fonts among our 100 fonts, and the bottom row show the 5 least interesting fonts (with interestingness scores decreasing from left to right). This is for our X2 case (please see the text for more details). See Figure 7 for all 100 fonts.

## ABSTRACT

While the problem of interestingness has been studied in various domains, it has not been explored for fonts. We study the novel problem of font interestingness in this paper. We first collect data of font interestingness in two ways, and analyze the data to understand what makes a font interesting. We then learn functions to compute font interestingness scores in two ways. We show results of rankings of fonts from the most to least interesting, and demonstrate applications of interestingness-guided font visualization and interestingness-guided font search.

**Index Terms:** Computing methodologies—Computer graphics—Perception

## 1 INTRODUCTION

While the problem of interestingness has been explored, in particular for images [16] and 3D shapes [20], it has not been explored for fonts. For fonts, while the attributes of fonts [26, 33] have been studied, the problem of font interestingness has been under-explored. In this paper, we aim to develop an understanding of the interestingness of fonts. Although fonts have a 2D representation, fonts are different from images. Images are usually colored representations of some scenes or objects, while fonts are grayscale, usually more sparse than images, and are the characterization of the letters and numbers used for text. In addition, we explore what features makes a font interesting, which are different from images in general.

Fonts have features that are worth studying. For example, sans-serif fonts have even width strokes and tend to more plain, so we hypothesize that these would be less interesting. On the other hand, serif fonts have thick and thin strokes, and we hypothesize that these would make such fonts more interesting. Script fonts are more fancy, elegant, personal and/or graceful, so we hypothesize that these would be more interesting.

To study the problem of font interestingness, we first collect data of font interestingness in two ways. First, we show participants one font at a time, and ask them to rate its interestingness. Second, we show participants pairs of fonts and ask them to judge which font of each pair is more interesting than the other.

We then analyze the collected data to understand what makes a font interesting. We ask more participants about various subjective features of the fonts, and check whether font interestingness is related to these subjective and qualitative features. We also compute various objective descriptors of the fonts, and check whether interestingness can be predicted by these descriptors.

We then learn font interestingness scores in two ways. First, with the data where we asked participants to rate each font, we learn a function that takes one font as input, and compute as output the font interestingness score. Second, with the data where we asked

participants to rate pairs of fonts, we specify a loss term and use gradient descent to find a function that also takes one font as input, and compute as output its interestingness score.

Finally, we show results of rankings of our fonts from the most to the least interesting in two ways, and analyze them to understand more about what makes a font interesting. We demonstrate the usefulness of our work with the applications of interestingness-guided font visualization and interestingness-guided font search.

In this paper, we make the following contributions:

- We are the first to study the problem of font interestingness to the best of our knowledge.

- We collect data of font interestingness in two different ways, and analyze the data to understand what makes a font interesting.

- We compute font interestingness scores and show that the concept of font interestingness can be learned.

- We demonstrate the potential uses of font interestingness through the applications of interestingness-guided font visualization and interestingness-guided font search.

## 2 RELATED WORK

Our work is inspired by previous works in fonts and interestingness (separately).

### 2.1 Font Attributes

O'Donovan et al. [26] developed interfaces for exploring large collections of fonts. They organized fonts using high-level descriptive attributes, such as attractive or not attractive, and showed fonts ordered by similarity relative to a query font. Our work is different in that we focus on studying one feature, compute interestingness scores for fonts, and explore what makes a font interesting. We believe that this particular feature is important for choosing fonts.

Wang et al. [33] took as input a set of predefined font attributes and their values to generate glyph images. Although they generated new fonts that their study participants found to be creative, they did not specify which fonts were creative, or analyze what makes them interesting or creative. Our work is different in that we compute a measure or score of how interesting a font is, and use this to understand more about what makes a font interesting.

Mackiewicz [24] examined the perceptions of fonts displayed on PowerPoint slides, where participants rated fonts on variables including "interesting". In this way, this previous work is closely related to our work. However, their work focused on designing presentations and is specifically for display on PowerPoint slides. In contrast, we focus on the problem of font interestingness in general,

analyze data to really understand what it means for a font to be interesting, and investigate whether a computational measure of font interestingness can be learned.

Researchers have shown that users may associate fonts with personalities [21, 25, 29, 30]. Our work is different in that we focus on studying one feature, and develop an understanding of interestingness for fonts, because we believe that this particular feature is important for choosing fonts. An interesting font makes the text fun and appealing, and makes it more likely to be read and enjoyed. Moreover, artists and designers could use interesting fonts to create attractive works that people are more likely to enjoy.

## 2.2 Research Problems related to Fonts

There are previous works studying various research problems related to fonts. These include font recognition [2], how fonts affect the emotional qualities (eg. more funny) of text [18], using font design as a tool for poster design [35], relations between the font of a brand and consumer perceptions of the brand personality [15], font specificity [28], and relations between fonts and reading speed [32]. In this paper, we study the novel problem of font interestingness.

## 2.3 Interestingness of various Media

Previous works have explored the interestingness of images [3, 6, 10, 14, 16, 31], videos [4, 7, 17, 34], 3D shapes [20], text passages [8], and interestingness measures for data mining [11]. These works show the importance of the research problem of interestingness. However, there has been no work in font interestingness, and we fill this gap in this paper.

## 2.4 Crowdsourcing

Previous works have used crowdsourcing to collect data from humans. Crowdsourcing has been used to collect style similarity data for clip art [9], fonts [26], and 3D models [22, 23]. Crowdsourcing has also been used to "extract depth layers or image normals from a photo" [12], and to "convert low-quality drawings into high-quality ones" [13]. In this paper, we use crowdsourcing to collect data of how humans perceive the interestingness of fonts.

## 3 COLLECTING DATA OF FONT INTERESTINGNESS

To study the problem of font interestingness, since there are no "right or wrong" answers to how interesting a font is, and different people may have different opinions, we take a human perception approach and collect data from humans.

We collected 100 fonts from an online library (fontlibrary.org), and collected data of font interestingness in two ways. First, we showed participants one font at a time and asked them to rate the font's interestingness on a Likert scale of 1-5. We call this our X1 case, since there is one input font per data sample. Second, we showed participants pairs of fonts and asked them to judge which font they perceive to be more interesting than the other. We call this our X2 case, since there are two input fonts per data sample. For the X2 case, we were inspired by previous works that ask users to select among triplets or pairs of items (e.g. clip art [9], fonts [26], and 3D models [22, 23]). We instruct users that an interesting shape is one that can attract or hold their attention in any way.

We use crowdsourcing as a method to collect data, and post the fonts on the Amazon Mechanical Turk platform. Each HIT (a set of questions on Mechanical Turk) starts with instructions for the participants. Since there are no correct answers to the questions, we did not filter out any responses that could be random (i.e. from users who answer randomly just to get paid). We encouraged users to be serious when answering the questions by specifying in the instructions that: "If you randomly choose your answers, your responses will not be taken, and you will not be paid." Also, each user can answer our questions only if their acceptance rate (i.e. as recorded

Figure 2: Top row: An example font (left) that scored low in aesthetics and interestingness, and an example font (right) that scored high in aesthetics and interestingness. Bottom row: An example font (left) that scored low in "serif-ness" and interestingness, and an example font (right) that scored high in "serif-ness" and interestingness.

by the requesters on Mechanical Turk) of their previous completed questions on Mechanical Turk is at least 90%.

For our X1 case, there are 50 fonts per HIT. The order of the fonts is chosen randomly. The users took between 1 and 4 minutes per HIT, and we paid $0.10 for each HIT. For each font, we collected data for 15 participants. For the X1 case, the font interestingness score is the average score given by the participants. For our X2 case, we generated 50 font pairs randomly for each HIT, by placing the fonts 1 to 100 randomly into 50 rows of 2. The users took between 2 and 5 minutes per HIT, and we paid $0.15 for each HIT. For each HIT (and thereby each font in this case), we collected data for 15 participants. For the X2 case, we have to perform the learning step in Section 5 to get the function that gives the font interestingness score. At the end of each HIT, we asked the participants to provide (by typing in a text box) their thoughts on how they decided how interesting a font is.

## 4 WHAT MAKES A FONT INTERESTING?

We analyze the collected data to try to understand what makes a font interesting. We first do this in a qualitative way. We collect additional data of how humans perceive the fonts according to some subjective features: creative, unusual, aesthetic, thin, serif, and italic. We also considered others features such as simple and fancy, but decided that these are too similar to the ones we have used already. For each feature and each font, we asked participants to provide a score on a 1-5 Likert-scale. For example, for "aesthetic", the participants would choose 5 if they strongly agree that the font is aesthetic. We also use the Amazon Mechanical Turk platform for this data, but note that the participants here are different from those in Section 3. There are also 50 fonts per HIT here, and the order of the fonts is chosen randomly. The users took between 1 and 5 minutes per HIT, and we paid $0.10 for each HIT. For each font, we collected data for 15 participants. The overall score of each feature for each font is the average score given by the participants. For each feature, we then correlated the scores for all fonts with the interestingness scores from our X1 case. We wish to see whether font interestingness is related to other qualitative features. The Pearson correlation coefficients for each feature are (in decreasing order): 0.8511 (creative), 0.7966 (unusual), 0.4794 (aesthetic), 0.3723 (serif), 0.2122 (italic), and 0.1425 (thin). The p-value for these are less than 0.05 and hence the correlations are significant. The only exception is for "thin" (with a p-value of 0.1573).

We discuss the results based on the correlation coefficients. Among the features we tested, "creative" has the highest correlation with interestingness. Hence the more creative a font is, then the more interesting it is. This is intuitive and is not a surprise, since a creative font tends to be fancy or appealing in some way, thereby making it interesting. A font that is more unusual is more interesting. An unusual font tends to be strange, weird, or stand out in some way, which makes it interesting. A font that is more aesthetic is more

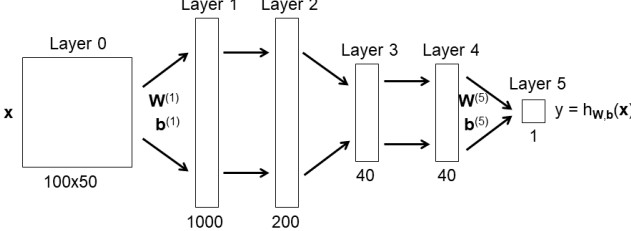

Figure 3: Our neural network with 6 layers. **x** is an input font and *y* is the font's interestingness score. The number of nodes is indicated for each layer. The network is fully connected. Note that this is for both our X1 and X2 cases. For the X2 case, we have two copies of this network for the batch gradient descent.

interesting. An aesthetic or beautiful font is good to look at and observe, which could be interesting. A serif font is more interesting. The wider and sharper parts of a serif font give it a characteristic look, which is interesting. An italic font is more interesting. It makes sense that slanted letters tend to be interesting. Finally, a font that is more thin is more interesting, but the correlation for this is not significant. In any case, thin letters are more like handwritten characters, so it makes sense that they could be more interesting.

To gain more insights into what makes a font interesting, we show some visual examples of fonts with varying aesthetics, serif, and interestingness scores (Figure 2). We found (from above) that more aesthetic fonts are more interesting. The figure shows one font with low scores for both aesthetics and interestingness. It is quite plain, and has the same letters for both capitals and non-capitals. There is also one font with high scores for both aesthetics and interestingness. It is a handwritten and cursive font. Furthermore, we found that serif fonts are more interesting. The figure shows one font with low scores for both serif and interestingness. It is thin and quite simple. There is also one font with high scores for both serif and interestingness. It is a bit cursive.

We then try to understand what makes a font interesting in a more quantitative way, by testing whether some quantitative 2D descriptors can be used to predict font interestingness. We learn a function that takes as input the 2D descriptors and compute as output the font's interestingness score from our X1 case. The 2D descriptors are: HoG (Histograms of Oriented Gradients) [5], SURF, SIFT, and the Sobel operator. Each descriptor is a histogram and we concatenate them into a single vector with a total of 1460 values. The function is a multi-layer neural network with fully-connected layers. We then perform 10-fold cross-validation and the resulting $R^2$ value is 0.46. The results are as expected as we did not think that a basic set of descriptors can predict interestingness, and the results show that the concept of font interestingness is complex.

## 5  LEARNING FONT INTERESTINGNESS

We learn font interestingness scores in two ways. First, our X1 case has one input font per data sample. We learn a function that takes as input a font and predict as output its interestingness score (Figure 3).

Second, our X2 case has two input fonts per data sample. With our pairwise fonts data, we follow the formulation in [19] which can take pairwise data and learn a function that computes a score for one font (which is also the network in Figure 3). Different from the usual supervised learning framework, we do not have the target values *y* that we wish to compute, as again our pairwise data are for pairs of fonts. Hence we take a learning-to-rank formulation [1, 27], and learn **W** and **b** to minimize this ranking loss function:

$$\mathscr{L}(\mathbf{W}, \mathbf{b}) = \frac{1}{2}\|\mathbf{W}\|_2^2 + \frac{C_{param}}{|\mathscr{I}_{train}|} \sum_{(\mathbf{x}_A, \mathbf{x}_B) \in \mathscr{I}_{train}} l_1(y_A - y_B) \quad (1)$$

where $\|\mathbf{W}\|_2^2$ is the $L^2$ regularizer to prevent over-fitting, $C_{param}$ is a hyper-parameter, $\mathscr{I}_{train}$ contains fonts $\mathbf{x}_A$ and $\mathbf{x}_B$ where the user specified that font *A* is more interesting than font *B*, $l_1(t) = \max(0, 1-t)^2$, and $y_A = h_{\mathbf{W},\mathbf{b}}(\mathbf{x}_A)$.

To minimize $\mathscr{L}(\mathbf{W}, \mathbf{b})$, we perform an end-to-end neural network backpropagation with batch gradient descent, and we follow the formulation in [19]. The forward propagation step takes each pair $(\mathbf{x}_A, \mathbf{x}_B) \in \mathscr{I}_{train}$ and propagates $\mathbf{x}_A$ and $\mathbf{x}_B$ through the network with the current $(\mathbf{W}, \mathbf{b})$ to get $y_A$ and $y_B$ respectively. Hence there are two copies of the network for each of the two cases *A* and *B*. We then perform a backward propagation step for each of the two copies of the network and compute these delta ($\delta$) values:

$$\delta_i^{(n_l)} = y(1 - y) \qquad \text{for output layer} \quad (2)$$

$$\delta_i^{(l)} = \left(\sum_{k=1}^{s_{l+1}} \delta_k^{(l+1)} w_{ki}^{(l+1)}\right)(1 - (a_i^{(l)})^2) \quad \text{for inner layers} \quad (3)$$

where the $\delta$ and *y* values are indexed as $\delta_{Ai}$ and $y_A$ in the case for *A*. The index *i* in $\delta$ is the neuron in the corresponding layer and there is only one node in our output layers. We use the *tanh* activation function which leads to these $\delta$ formulas. Note that due to the learning-to-rank aspect, these $\delta$ are different from the usual $\delta$ in the usual neural network backpropagation.

We now compute the partial derivatives for the gradient descent. For $\frac{\partial \mathscr{L}}{\partial w_{ij}^{(l)}}$, we split this into a $\frac{\partial \mathscr{L}}{\partial \|\mathbf{W}\|_2} \frac{\partial \|\mathbf{W}\|_2}{\partial w_{ij}^{(l)}}$ term and $\frac{\partial \mathscr{L}}{\partial y} \frac{\partial y}{\partial w_{ij}^{(l)}}$ terms (a term for each $y_A$ and each $y_B$ computed from each $(\mathbf{x}_A, \mathbf{x}_B)$ pair). The $\frac{\partial \mathscr{L}}{\partial y} \frac{\partial y}{\partial w_{ij}^{(l)}}$ term is expanded for the *A* case for example to $\frac{\partial \mathscr{L}}{\partial y_A} \frac{\partial y_A}{\partial a_i} \frac{\partial a_i}{\partial z_i} \frac{\partial z_i}{\partial w_{ij}^{(l)}}$ where the last three partial derivatives are computed with the copy of the network for the *A* case. The entire partial derivative is:

$$\frac{\partial \mathscr{L}}{\partial w_{ij}^{(l)}} = w_{ij}^{(l)}$$
$$+ \frac{2C_{param}}{|\mathscr{I}_{train}|} \sum_{(A,B)} max(0, 1 - y_A + y_B) \, chk(y_A - y_B) \, \delta_{Ai}^{(l+1)} a_{Aj}^{(l)} \quad (4)$$
$$- \frac{2C_{param}}{|\mathscr{I}_{train}|} \sum_{(A,B)} max(0, 1 - y_A + y_B) \, chk(y_A - y_B) \, \delta_{Bi}^{(l+1)} a_{Bj}^{(l)}$$

There is one term for each of the *A* and *B* cases. $(A, B)$ represents $(\mathbf{x}_A, \mathbf{x}_B) \in \mathscr{I}_{train}$ and all terms in the summation can be computed with the corresponding $(\mathbf{x}_A, \mathbf{x}_B)$ pair. The *chk*() function is:

$$chk(t) = 0 \qquad \text{if } t \geq 1 \quad (5)$$
$$= -1 \qquad \text{if } t < 1 \quad (6)$$

For each $(A, B)$ pair, we can check the value of $chk(y_A - y_B)$ before doing the backpropagation. If it is zero, we do not have to perform the backpropagation for that pair as the term in the summation is zero. The partial derivative for the biases is derived similarly.

The batch gradient descent starts by initializing **W** and **b** randomly. It then goes through the fonts for a fixed number of iterations, where each iteration involves taking a set of data samples and computing the partial derivatives. Each iteration of batch gradient descent sums the partial derivatives from a set of data samples and updates **W** and **b** with a learning rate $\alpha$ as usual:

$$w_{ij}^{(l)} = w_{ij}^{(l)} - \alpha \frac{\partial \mathscr{L}}{\partial w_{ij}^{(l)}} \quad (7)$$

$$b_i^{(l)} = b_i^{(l)} - \alpha \frac{\partial \mathscr{L}}{\partial b_i^{(l)}} \quad (8)$$

## 6  RESULTS AND EVALUATION

We show results of the 5 most interesting fonts and the 5 least interesting fonts for our X1 and X2 cases (Figures 1 and 4). In Figure 1, the 5 most interesting fonts include some handwritten fonts, some aesthetic fonts, and an unusual font consisting of QR codes. The 5

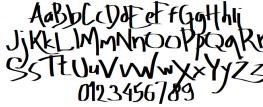

Figure 4: The top row shows the 5 most interesting fonts among our 100 fonts, and the bottom row show the 5 least interesting fonts (with interestingness scores decreasing from left to right). This is for our X1 case (please see the text for more details). See Figure 8 for all 100 fonts.

least interesting fonts include some sans-serif fonts, some italic fonts, some simple fonts, and some fonts with thin characters. In Figure 4, the 5 most interesting fonts include some handwritten fonts, and some unusual fonts. The 5 least interesting fonts are mostly simple fonts. They include some non-italic fonts, some serif fonts, and some sans-serif fonts. Comparing the X1 and X2 cases, the 5 most interesting fonts for both cases are visually similar, and the 5 least interesting fonts for both cases are also visually similar. Therefore, although the X1 and X2 cases collect data differently, the visual results are similar. Moreover, we note that our qualitative analysis of what is an interesting font is subjective, and we encourage the reader to observe the fonts for themselves.

After the participants answered a set of questions, they provided their thoughts on how they decided how interesting a font is. For the X1 case, they said: "handwriting looks more beautiful and interesting", "unusualness, strangeness is interesting", "look for curved parts, thin parts", "plain and straight not interesting", "try to compare with previous fonts", "could have more than 5 options", and "sometimes some letters look better, and some are less interesting, so have to balance them". For the X2 case, the participants said: 'look at details of curves and edges, or whether tips of strokes bend back versus are more straight", "if both interesting, it's difficult", "sometimes both plain, then it is the same", "more cute, more pretty is better", "thin or italics more interesting", "strange/unusual makes it more interesting too", "fancy is more interesting", "look for more cursive or handwritten letters", "capital letter is less interesting than non-capital letter", and "special characters are more interesting". We note that for the X2 case, in general, the words "both" and "more" are often used, which makes sense for comparing between fonts.

In the introduction, we made the hypotheses that sans-serif fonts are less interesting, serif fonts are more interesting, and script fonts are more interesting. These hypotheses are correct, as we found in Section 4 that aesthetic and serif fonts are more interesting. Moreover, the participants' words (above) agree with these hypotheses.

We provide some parameters used in our method. The hyperparameter $C_{param}$ is set to 1000. We initialize each weight and bias in $\mathbf{W}$ and $\mathbf{b}$ by sampling from a normal distribution with mean 0 and standard deviation 0.1. We go through all the fonts 100 times or more for the network to produce reasonable results. For each iteration of the batch gradient descent, we choose between 50 and 100 data samples for $\mathscr{I}_{train}$. The learning rate $\alpha$ is set to 0.0001. The training step can be done offline. For example, 100 iterations of batch gradient descent for one font takes about 3 seconds in MATLAB. This runtime scales linearly as the number of fonts increases.

We describe the accuracy of the learning method. For the X1 case, we perform a 10-fold cross-validation, and the resulting $R^2$ value is 0.71. For the X2 case, after the training step, we can use the neural network to compute an interestingness score for each font. We take all font pairs in the data and perform 10-fold cross-validation. For each pair $(\mathbf{x}_A, \mathbf{x}_B)$, we compute $y_A$ and $y_B$ with the trained network, and then predict the font with the higher score to be more interesting. The percentage of samples where the participant interestingness response is predicted correctly is 76.3%.

Figure 5: Interestingness-Guided Font Visualization. Instead of showing all 100 fonts, we choose a subset of fonts as one way to visualize them.

## 7 APPLICATION

We demonstrate the potential uses of the font interestingness concept with some interestingness-guided applications.

We show an application of interestingness-guided font visualization. The idea is to choose a subset of fonts to visualize the whole set of fonts. It would be useful to visualize all the fonts with a smaller subset, both to understand what types of fonts are in the set, and to choose a font to use from a smaller subset. One way to do so is to take the fonts ranked according to interestingness, and choose one for every k fonts. For our X1 case, we tried this for k=5 and 10, but there were too many fonts chosen which did not look good. We then decided to do this with k=20 (Figure 5), to get a subset of five fonts. Among this subset, the first font is cursive, then they are more and more simple or plain looking. From the second font, they alternate between serif and sans-serif fonts. There are a variety of fonts: there are one handwritten, two italic, two non-italic and one bold font. The only main difference from the set of 100 fonts is that these do not include some unusual font, but there is a good aspect to this, as an unusual font is not likely to be used in practice. We performed a test and posted the subset of fonts on Amazon Mechanical Turk, and asked 15 participants to give a Likert-scale rating of 1-5 for these statements: "It is useful to visualize the set of fonts this way" (in this case, we also showed the set of all fonts) and "You can choose a font to use from these". The average rating for the first statement is 4.2, and for the second statement is 4.5.

We show another application of interestingness-guided font search. Figure 6 shows a query font, the top-4 results from searching with interestingness scores, and then the top-4 results from searching with other 2D descriptors. The query font has high interestingness, and the first row has interesting fonts. The other rows have fonts that are not high in interestingness, except for one (the third font in the second row). This shows that if a user wants to search for a font according to interestingness, applying our font interestingness scores would be useful.

## 8 DISCUSSION, LIMITATIONS, AND FUTURE WORK

We have investigated the novel problem of font interestingness, and started to develop a computational understanding of this concept. We demonstrate in this paper that this is a worthwhile problem to study, and hope that our research will inspire more work.

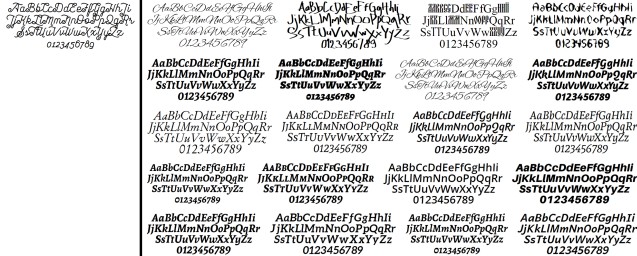

Figure 6: Interestingness-Guided Font Search. The top left font is the query. Each row shows the top-4 results from searching based on our interestingness scores, HoG, SIFT, SURF, and Sobel descriptors respectively.

One limitation is that we currently have 100 fonts (although it took much time to prepare these fonts). For future work, we may gather more fonts and collect more data.

We currently have only black-colored fonts with a white background. For future work, it is possible to have colored fonts, and/or colored backgrounds, or even decorations on the characters, such as in the recent Google doodle for the December holidays.

Currently, the learned function is a neural network that is relatively simple. However, the learning itself is not our contribution, but the goal was to show that the concept of font interestingness can be learned. We may explore more complex functions in future work.

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

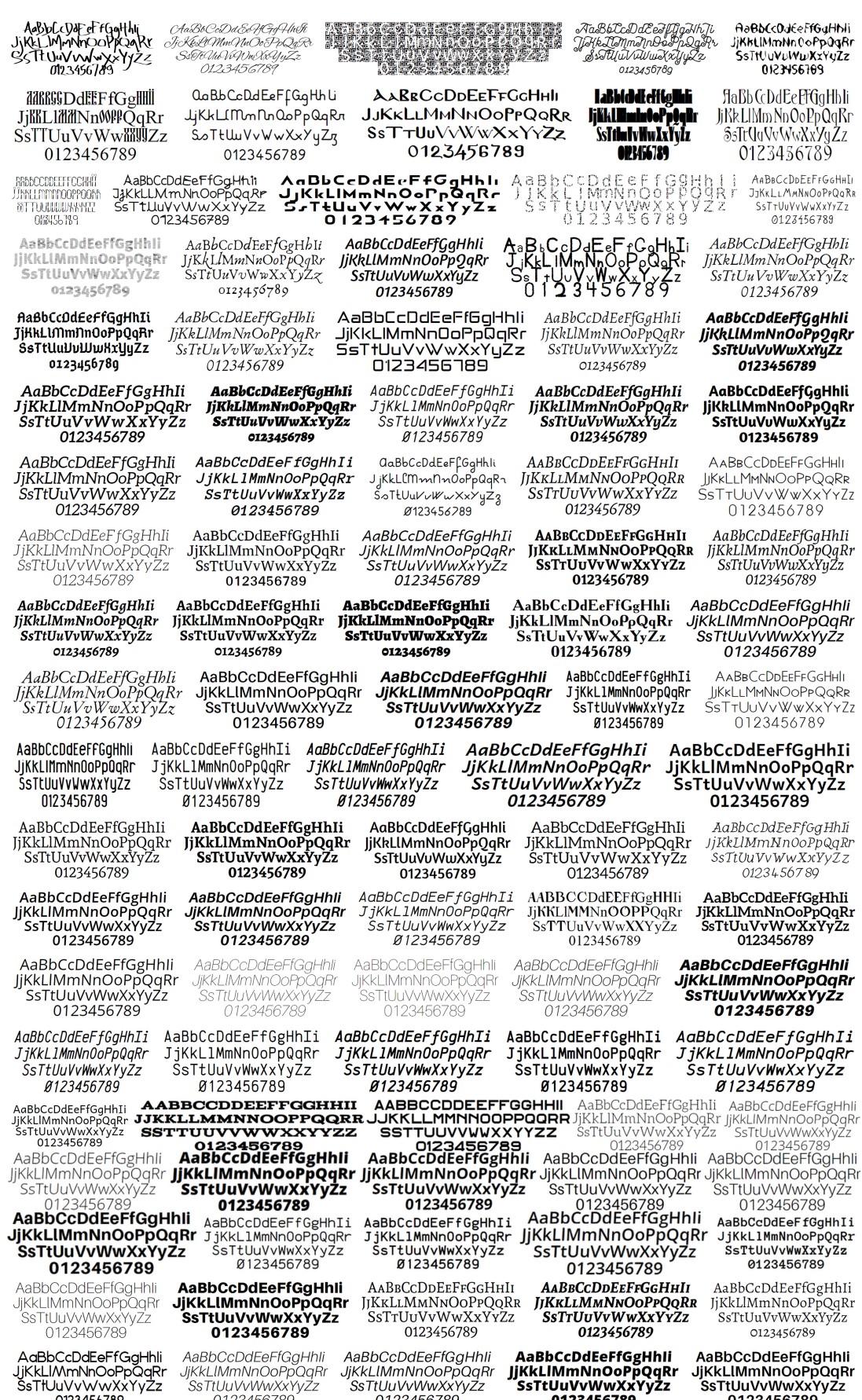

Figure 7: The 100 fonts are ranked from most to least interesting (from top left, and left to right in each row). This is for our X2 case. Please see the text for more details, and please zoom in to better see the fonts.

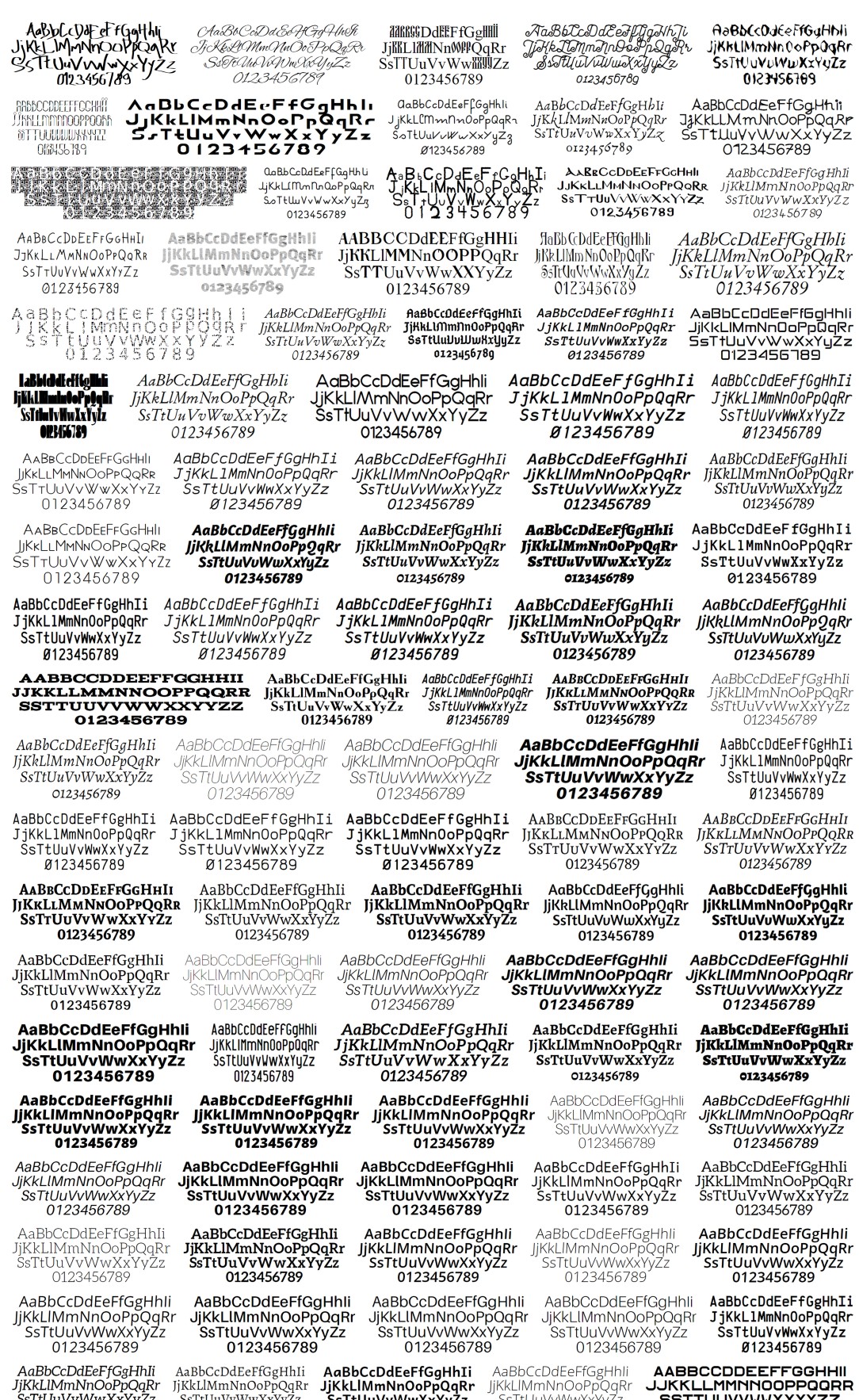

Figure 8: The 100 fonts are ranked from most to least interesting (from top left, and left to right in each row). This is for our X1 case. Please see the text for more details, and please zoom in to better see the fonts.

