# OpenReview forum: "The Interestingness of Fonts"
_graphicsinterface.org/Graphics_Interface/2023/Conference — Submitted to GI 2023_

### Official Review · Reviewer_mup2 · 2023-01-13
**Mildly interesting; weird study**

**Rating:** 5
**Confidence:** 3

**Review:**

The paper studies participants' perception of fonts' interestingness and tries to replicate the ranking they get from user study data by fitting a fully-connected neural network, with some success.

The topic itself is rather curious, and perhaps, although unlikely, might have an actual application. However, I was underwhelmed by the study itself: somehow, instead of a variety of font-specific geometric properties, like line style, serif style, how constant is the line width, slant, etc., they only looked at the binary flags 'serif', 'cursive', and 'italic'. This is a very limited scope.

 Some of the 'features' the authors studied were, in my view, completely irrelevant, such as "creative", "unusual", or "aesthetic". These qualities are 1) subjective, 2) irrelevant to fonts, are probably correlated with many 'interesting' classes of objects, and 3) the first two can be simply considered synonyms of the word 'interesting'. Such analysis not only does not provide any useful insight into the 'interestingness' of fonts, but is mostly misleading.

The trained networks are rather standard, although I find the experiment with using 2D descriptors good and useful.

Finally, although this does not influence the score I give, I find paying $3/hour on MTurk immoral.

In general, I am on the fence: I find the paper mildly interesting, but cannot understand the study design choices the authors made.

---

### Official Review · Reviewer_Dp5Z · 2023-01-13
**novel topic? invalid methodology**

**Rating:** 2
**Confidence:** 3

**Review:**


This paper studies the interestedness of fonts through a user-based experimental framework.

Unfortunately, even if one agrees to the existence of the concept of interestingness of fonts, which I personally do not really submit to, there are major problems with the methodology.

First, the user studies. The choice of having user queries with one and two fonts is not explained properly and to me does not make sense.

Second, if the problem is determining the most interesting font, this requires only analyzing the user study, I do not understand very well the learning part. If the idea is to extract some sort of predictor for font interestedness, this part of the paper is very unclear and improperly evaluated. From what I understand, the authors use all 100 fonts in the training data so it is not clear how the validation was performed, Section 5 is very unclear to me, Many details are very unclear, including basic things like the architecture in Figure 3.

The authors list 3 contributions:

1) the first to study the problem of font interestingness --> ok, not sure I understand why this is a contribution

2) We collect data of font interestingness in two different ways --> both ways are pretty trivial and not explained properly

3) We compute font interestingness scores and show that the concept of font interestingness can be learned. This part is unclear and the methodology appears to have major flaws starting with the fact that there seems to be no testing data applied outside of the training data.

Furthermore, I do not understand the application section - both applications presented are not clear at all.

---

### Official Review · Reviewer_Bn34 · 2023-01-14
**This paper builds a model of font "interestingness" based on ratings given by participants. Although the model works fairly well, the use-cases for the model could be more convincing.**

**Rating:** 5
**Confidence:** 4

**Review:**

This paper builds a model of font "interestingness" based on ratings given by participants in two studies. The first study asks participants to rate how interesting a font is on a Likert scale. The second study asks participants to choose which of the two presented fonts is more interesting. The authors build a predictive model of interestingness based on their participant's ratings.

The ability to rank fonts based on "interesting" could be a fun way
to filter and order fonts. The authors perform both quantitative and qualitative analyses that contribute to our understanding of how people perceive fonts.

The authors could better motivate how a measure of interestingness fits into how fonts are typically used in practice, particularly in graphic design. In other words, how might interestingness fit into "best
practices" of when to use a particular font? Depending on the context, a simple font might be more visually appealing.

In section 2.1, the authors claim that "an interesting font makes the text fun and appealing, makes the text more likely to be read and enjoyed". Is there evidence for this? In a typical document, a non-standard font can be harder to read.

In section 4, Their qualitative analysis investigates which font attributes (creative, beautiful, thin, etc) are correlated with "interesting".  I would also be interested in how measurable font characteristics -- such as curvature, line thickness, slant, ascent, descent, etc -- be associated with interestingness.

In section 4, how might serif, which refers to whether a font has
decorative small strokes (like Times New Roman), be a subjective font attribute? Did you define font terminology, such as serif, at the beginning of the experiment?

In section 4, you describe a multi-layer neural network that you train on font descriptors. Why use this model? Did the authors consider performing a statistical analysis to see which fonts, or which font characteristics, are significantly more interesting than others?

In section 7, the survey question "It is useful to visualize the set of
fonts this way" is a leading question that likely overestimates the rating of the system.

---

### Meta-Review · Area_Chair_s3xP · 2023-01-17

**Recommendation:** 4
**Confidence:** 4

**Metareview:**

In general, the reviewers agree that the problem is rather interesting on its own, although some reviewers doubt its practical applications. More importantly, however, the reviewers point out the issues with the user studies, such as 'serif' being a subjective attribute, capturing correlations between words ('creative' and 'interesting') as opposed to the font features, as well as the design of the study itself (1 font vs 2 fonts). Finally, some of the sections are missing critical details and are not explained throughly.

As a result, the reviewers are recommending rejection.